# Investigation of Sterile Mining Dumps Resulting from Ore Exploitation and Processing in Maramures County, Romania

Ioana Andreea Petrean [1,*], Valer Micle [1,*] and Marin Şenilă [2]

1 Department of Environment Engineering and Entrepreneurship of Sustainable Development, Faculty of Materials and Environmental Engineering, Technical University of Cluj-Napoca, 400114 Cluj-Napoca, Romania
2 INCDO INOE 2000, Research Institute for Analytical Instrumentation, 400293 Cluj-Napoca, Romania
* Correspondence: ioana.petrean@imadd.utcluj.ro or ioanna_petrean@yahoo.com (I.A.P.); valer.micle@imadd.utcluj.ro (V.M.)

**Abstract:** Abandoned sterile dumps can be a significant source of environmental pollution, therefore the distribution of trace elements throughout mining is vital. Monitoring environmental factors in closed mining perimeters aims to track the quality of discharged waters in the emissary and assess acid mine drainage, the quality of the soil and vegetation, stability, and the condition of the land surfaces within the perimeter of the sterile deposits. One of the primary sources of land, water and air pollution is sterile mining dumps. Knowing the source of pollution is the first step in adequately managing the affected areas. This paper investigates the physical–chemical properties and the concentrations of heavy metals in sterile dumps resulting from mining. We studied one sterile dump that was the result of ore processing and whose surface was covered with abandoned mixed ore concentrate (Pb, Zn), located in the Băiuț mining area (Romania), and a second sterile mining dump that was the result of exploration and exploitation work in the Ilba mining area (Romania). In order to determine the physicochemical characteristics of the studied sterile dumps and to determine the concentration of heavy metals, 27 sterile samples and one soil sample were taken from the Băiuț dump. Additionally, 10 sterile samples and one soil sample were collected from the Ilba dump. *Aqua regia* extractable concentrations of heavy metals were determined by inductively coupled plasma optical emission spectrometry. At the same time, using a portable XRF, we analyzed selected samples from each site for total metal concentrations. Furthermore, from each site, one sample was analyzed by FT–IR spectrometry. The pH values in both sterile dumps were highly acidic ($\leq 3.5$) and the content of heavy metals was generally increased.

**Keywords:** anthropogenic pollution; heavy metals; sulfides; ICP–OES; portable XRF; FT–IR; remediation





## 1. Introduction

To maintain ecological balance and to improve natural factors, the quality of life, and the conditions that may affect human and animal health, the protection of the environment is a crucial part of sustainable social development [1]. Heavy metals should be present in low concentrations in most cases, but significant input and flow of heavy metals into the biosphere is a major global problem [2]. Industrial activities such as mining operations, smelting, and refining processes are one of the leading causes of heavy metal contamination worldwide [3], caused by waste residue enrichment and inappropriate environmental management, leading to environmental pollution determined by the transformation and migration of pollutants [4].

Mining extraction waste, tailing ponds, and mining sterile dumps are the primary sources of water pollution in rivers, responsible for the extinction of fish, flora, and fauna [5]. Heavy metal (Pb and Cr) contamination risk in sediments from Hara Biosphere Reserve in southern Iran due to anthropogenic activities is associated with adverse biological effects [6]. Concentrations of heavy metals (Cr, Ni, Cu, As, Cd, and Pb) in the Buriganga

River, Bangladesh, indicated progressive deterioration of the surface sediment and showed a considerable to very high potential ecological risk [7].

Heavy metals found in mining wastes are quickly discharged into various environmental media, possibly causing harm to human health and ecosystems [5]. Human health risk assessment of heavy metals (As, Hg, Ni, Pb, Zn) in water samples used for drinking, domestic, fishing and irrigation purposes associated with abandoned barite mines in Cross River State, southeastern Nigeria, indicated an unacceptable risk for non-carcinogenic adverse health effects [8]. The potential environmental health risk of the heavy metal (Mn, Cu, Zn, As, and Cd) concentration of surface water from an abandoned mine in Smolnik creek in Slovakia through ingestion and dermal contact showed an unacceptable cancer risk threshold for adults and children [9]. The exploitation of Pb-Zn ore and other associated metals in the Artana mine (Novo Brdo) in Kosovo caused the Marec river and the environment in its vicinity in general to become polluted. [10]. The pollution of the urban environment in the Mitrovica region in Kosovo, caused by the Trepça Mine, is a severe and permanent concern due to the impact of heavy metal concentrations in the industrial discharge water [11].

The mining and processing of metal ores may be a substantial source of environmental pollution due to the increased occurrence of heavy metals to dangerous levels, compromising environmental quality and ecosystem functions [12]. The overall hazard index for soil and plants in the region of Shanono and Bagwai gold mining environments, Nigeria, was found to be greater in children compared to adults, while that of water was higher in adults compared to children; moreover, 6 out of 1 million children and 7 out of 1 million adults may develop excess cancer in their lifetime as a result of heavy metal exposure in the area [13]. The heavy metals analyzed in water, fish, nails and scalp hair in children between 5 and 10 years in the Migori gold mining belt in Kenya indicate that the children are exposed to high health risks associated with consuming contaminated water and fish from the rivers in the area [14].

Metalliferous sterile mining materials cause surface or groundwater pollution, offsite contamination via aeolian dispersion, water erosion, and bioaccumulation and uptake by vegetation in food chains in cases of poor mining closure management, posing severe risks to human health and agricultural activity [12,15–18]. Heavy metal contamination in drinking water from shallow groundwater wells in Ubon Ratchathani province, Thailand, suggested increased cases of non-carcinogenic and carcinogenic health defects among locals [19]. Dust from the mining dumps, flotation processes, and quarries loaded with heavy metals is displaced by the currents of the atmosphere, creating a pollution factor and a source of health risk due to the high level of toxic substances (Pb, Cd, Zn, Cu, Hg, Ni, Cr, Se, As) [20]. Moss and lichen samples used to monitor air quality exposed at different distances from Slovinky tailing pond (Slovakia) showed high Pb, Zn, Ni and Fe values in samples exposed at a 200-m distance [21].

Some heavy metals are known human carcinogens (As, Cr, Pb), whereas others are systemic non-carcinogens, causing harmful effects throughout the body even at low exposure levels [22]. Heavy metals (Pb, Cd, Zn, Cu, and Cr) from the surface soils of public parks in southern Ghana expose humans to health risks and tend to accumulate beyond the acceptable limit, demonstrating the need for close monitoring [23]. In a Pb–Zn mining area in Hunan province (China), the average concentrations of As, Pb, Cd and Zn in soils exceed the limits of the Chinese National Soil Environmental Quality Standard III, and Cd showed a very high potential ecological risk [24]. High concentrations of heavy metals, mainly Pb, were found in soil samples from a gold mining area in northern Atbara (Dar-Mali locality), River Nile State, Sudan [25].

The inadequate management of mining-related wastes has become a severe concern for the environment and human health [26]. Heavy metal pollution in the soils around the Kapan mining area in Armenia showed increased human health risks through different exposure pathways [26]. Soils in Armenia, near the Shamlugh copper mine, Chochkan tailing dump, and ore transportation road, have posed serious health risks to the biological

health of soil and humans and are dangerous for agricultural production due to Cu, Pb, As, Ni, Zn and Co concentrations [27]. The health risks of heavy metals in the soil of Nigerian mining districts revealed that adults are more vulnerable, with ingestion being the most significant contributor to excess lifetime cancer risk, followed by cutaneous pathways [28].

Mining creates severe environmental damage by altering the terrain, deteriorating land areas, eradicating wildlife, and clearing vegetation [29]. The infrastructure of sterile mining dumps and tailings deposits causes irreversible environmental, social, and human life losses, making it an international concern [30]. In many of the world's mining sites, waste mining material is deposited as spoil heaps in nearby narrow valleys and on steep slopes of small streams. The physical stability of the sterile mining dumps with regard to constructive faults that may produce overflows or a collapse and their chemical stability, such as the control of acidification reactions and avoidance of contamination of the surrounding ecosystems, are the main concerns regarding sterile mining deposits [31,32].

Instability phenomena such as incorrect dimensioning or non-compliance with the geometrical elements of the slopes, too much inclination resulting from incorrect geometrical configuration, or the accumulation of excess sterile material exceeding the bearing capacity, etc. cause problems with the stability of sterile mining deposits [33]. Mining risk factors for sterile mining dumps include potential detachments and landslides caused by increased slope or congestion of existing under–crossings and caustic mining concentrate [34,35]. The diminished stability of aggregate makes the soil more vulnerable to leaks, continual erosion, and crusts (frequently connected with decreased soil aeration and permeability) [36]. With leaks or infiltrated water, the clay fraction can carry adsorbed nutrients or pollutants along the soil surface or down into the soil profile [37].

Clogging of sewers under sterile dumps produced by material entrained from the riverbed, slopes, branches, garbage, and so on can result in vast discharges of sterile material and water downstream via the accumulation of water upstream [38]. Due to the inconsistency of the materials in the mine waste deposits and non-compliance with geotechnical measures, there is a potential risk of blockage of some sections of the local hydrographic network due to the high level of materials accumulated in the minor riverbeds, creating favorable conditions for flooding [20]. Mining waste deposits are highly susceptible to erosion through collapses or mass slides [33,35].

The main purpose of this study was to identify the chemical status of the sterile mining dumps in Maramures county, NW of Romania, by analyzing a sterile dump from Băiuț resulting from ore processing and a sterile dump from Ilba resulting from exploration and exploitation, which have been significantly impacted by mining activity in the Maramures metallogenic area. The possible impact of heavy metals on the environment and on human health is the reason for analyzing the state of degradation of the sterile dumps in Maramures county. In addition, the present study contains images from the on–site field visits conducted to create a complete picture of what remained following mining activities in the studied areas.

## 2. Materials and Methods
### 2.1. Areas of Study and Geologic Data
#### 2.1.1. Băiuț Mining Perimeter

The Bloaja sterile dump (lat. 47°58′14.8″ N and long. 23°98′78.2″ E), as a result of ore processing, contains sterile mining and abandoned mixed ore concentrate (Pb, Zn). The study area is located in Romania on the territory of Băiuț commune, in the county of Maramureş. The mountains in the area are made up of volcanic rocks and canton deposits of gold, silver, zinc, lead, and copper [39,40]. The relief of the Baiuț mining perimeter is characterized by rugged mountains, streams with longitudinal and steep profiles, numerous thresholds and waterfalls, and the creeks in the region generally have modest flow, being fed by precipitation and snowmelt, as well as certain subsurface sources [41]. The precipitation ranges from 950 mm to 1380 mm. The Lăpuș River collects all the rivers in the area; it is the longest river in Maramureş County (114 km) with a basin area of 1820 km$^2$ and

the main right tributary of the Somes River, which is almost the unique collector of the southern slopes of the Gutâi, Țibleş and Lăpuș Mountains. The average drain is around 500 mm, while the average debit at the Răzoare is 10 m$^3$/s. The maximum debit rate can be 580 m$^3$/s, and the autumn minimum debit rate can be 0.35 m$^3$/s [42]. The dominant soils of the Băiuț area are represented mainly by Lithosols, Dystricambosol, and Alluviosols in patches by Eutricambisol, Luvosol, Andosol, and Regosol [41,43].

Băiuț's epitermic mineralization is found in the Văratec Mountains; the majority of them have a philonian shape; gold predominates in the upper part of the deposit and silver in the lower parts [39]. Cisma ore from the metallogenetic field of Băiuț is rich in Cu, with the presence of precious metals; in the Breiner-Băiuț epithermal ore deposit, sphalerites appear with chalcopyrite, pyrite, galena, tetrahedrite, iron oxides, quartz and carbonates [44,45].

Mining operations in Băiuț were first recorded in 1315 during the medieval period [46], and together with the Baia Mare mining region, it is one of the most critical regions for metal exploitation. The Baiuț mining perimeter has been an important exploitation center for the mining of base metals since its establishment in 1769 [47]. The objective of mining activity in the Băiuț area was the exploitation in filons of base metal and Au-Ag ores situated in Paleogene and Sarmatian clastic rocks, neogenous volcanic reserves, or subvolcanic bodies [40,47]. The flotation ore processing plant (Figure 1) at Băiuț had a preparation capacity of 20,000 tons of ore and an annual production of about 500 kg of fine gold during its operation following investments between 1931 and 1940 [48].

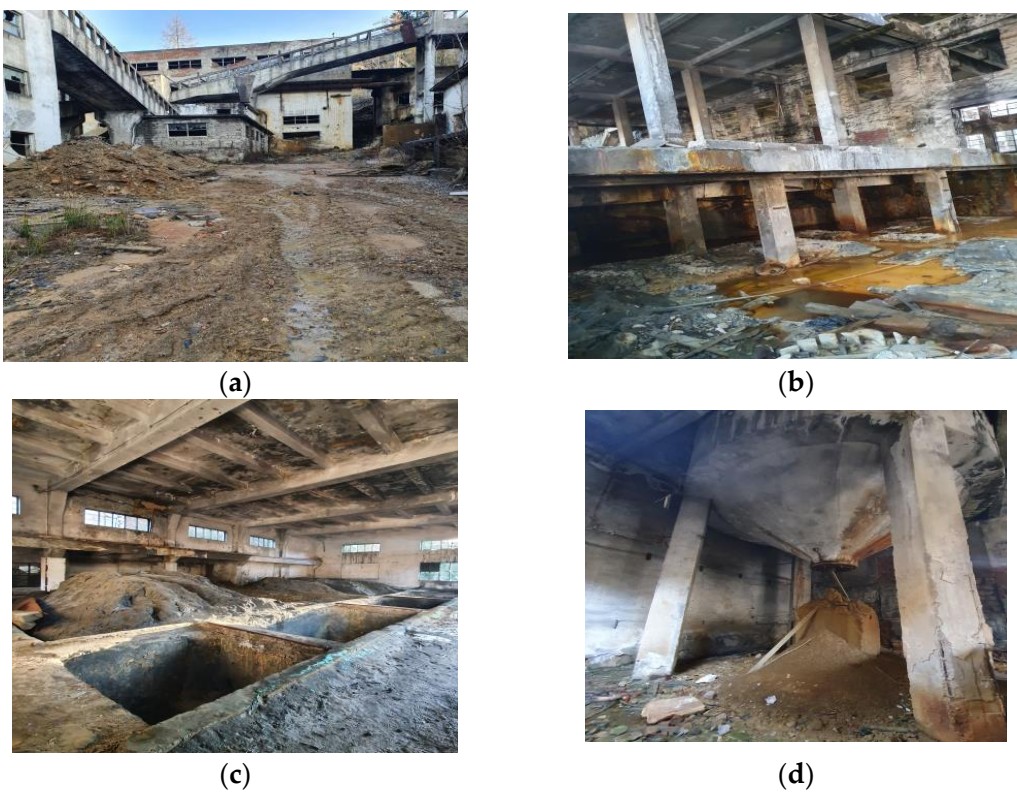

(a)　　　　　　　　　　(b)

(c)　　　　　　　　　　(d)

**Figure 1.** The ruins of the flotation treatment plant Băiuț with abandoned mining concentrate and sterile, Maramureș county, Romania (photos: I.A.P.): (**a**) Interior courtyard with sterile leaks; (**b**) Interior of building ruins of the treatment plant; (**c**) The platform of silos filling with mining concentrate; (**d**) Silo hopper and sterile mining.

The exploitation of sulfidic ore deposits from Băiuț closed in 2007 [39]. Although work ceased in the Băiuț area after the closure of the mines and the flotation plant, numerous sterile deposits as well as acidic mine water, which is continuously poured into the Tocila and Băiuț rivers, still remain [49]. In the area, it was estimated that almost 1,300,000 m$^3$ of sterile coarse-grained material and 900,000 m$^3$ of sterile flotation waste were deposited in ponds and the Lăpuș River (Figure 2), which now act as heavy metal distribution networks [47].

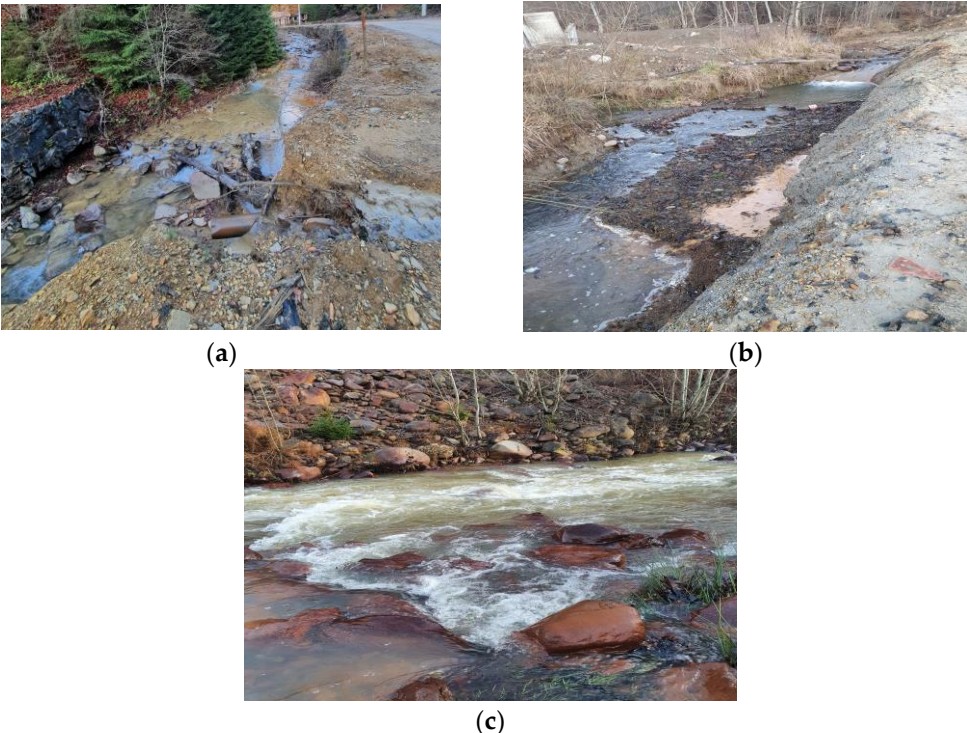

(**a**)         (**b**)

(**c**)

**Figure 2.** The flow of mine water in Băiuț valley, the Bloaja Tailing Valley and the collector Lăpuș River, Băiuț-Văratec metalliferous mining area, Maramureș county, Romania (photos: I.A.P.): (**a**) Băiuț valley; (**b**) Bloaja Tailing Valley; (**c**) Lăpuș River.

On the platform of the studied site, in the lower Bloaja dump (Figure 3), several thousand tons of mixed ore concentrate (Pb, Zn) were abandoned [47].

The concentration of stored pyrite (over 50,000 tons) causes acidification of the soil in neighboring areas and corrodes the inverted pump system that evacuates rainwater from the platform of the dump. As a result, four large holes have formed at the dump site [50]. As the dump is chemically and physically unstable and its storage capacity has been reached, concentrated and sterile water has flowed from the slope and leached into the surrounding soil and then into the Lăpuș River (Figure 4).

### 2.1.2. Ilba Mining Perimeter

The Ilba exploitation perimeter (lat. 47°75′76.3″ N and long. 23°39′51.5″ E) is located in the southwestern extremity of the Oaș-Gutâi eruptive chain. The relief of the Ilba-Cicîrlău area consists of cones, craters, lava plateaus and volcanic agglomerates, products of Neogene volcanic eruptions belonging to the western frame of the Eastern Carpathians [51,52]. Annual rainfall is 700–800 mm. The soils specific to the Ilba area are represented by brown argiloiluvial soils, brown podzolic soils, stagnosols and alluviosols. The mountain hydrographic network consists of two valleys: the Cicîrlău Valley and the Ilba Valley [46]. The myocene hydrothermal deposit Ilba, the largest of the Maramures volcanic orogen's northern group, was formed by a cluster of lava and volcanic agglomerates, in which basaltoid andesites dominate [40,46]. Ilba's mineralization has a predominantly polymetal-

lic, gold-copper or pyrithos character [51]. The deposit of Ilba is one of the important non-ferrous ore deposits in the Baia Mare mining basin. Mining at Handalu Ilbei began in ancient times; in the Middle Ages, there were particularly active exploitations, and in the area of the Colbului Valley, there were also ore-crushing stamps and furnaces [46].

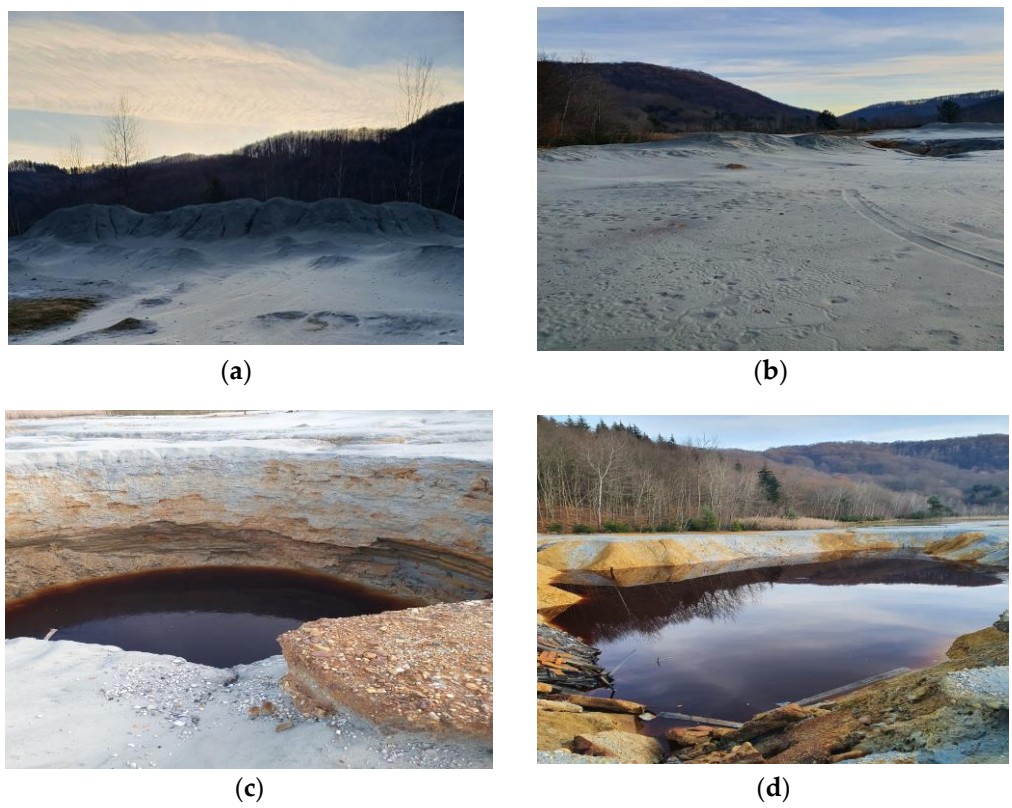

**Figure 3.** Mixed ore concentrate on the corroded platform of Bloaja sterile dump, Băiuț area, Maramures county, Romania (photos: I.A.P.): (**a**) Piles of ore concentrate; (**b**) Mining concentrate dispersed by atmospheric agents on the surface of the sterile dump; (**c**) Huge circular hole on the surface of the sterile dump caused by mining concentrate; (**d**) Huge and wide hole on the surface of the sterile dump filled with acid water.

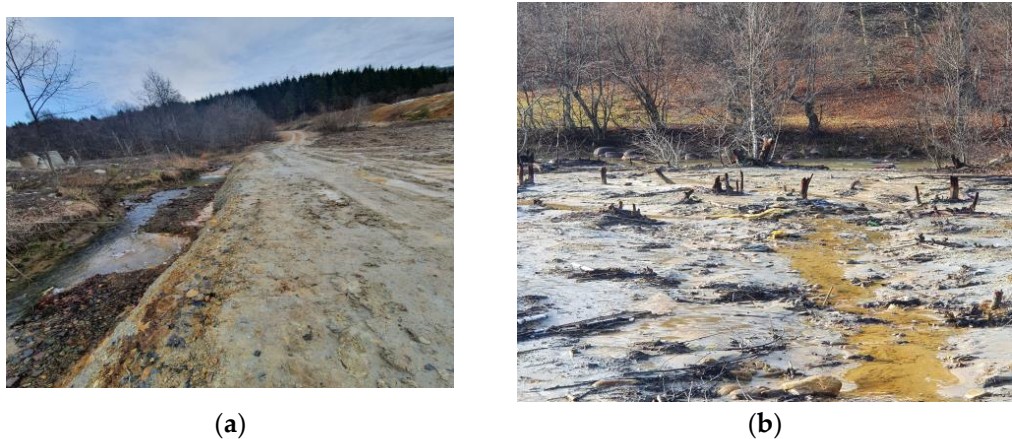

**Figure 4.** Leaks from the Bloaja dump directed on the gutter and in the Lăpuș River, Băiuț area, Maramures county, Romania (photos: I.A.P.): (**a**) Leaks in the gutter downstream of the sterile dump; (**b**) Concentrate and acid water flowing into the Lăpuș River.

The sterile material from the Ilba sterile mining dump (lat. 47°75′66.3″ N and long. 23°39′35.9″ E) is the result of exploration and exploitation and was gravitationally deposited in the dump and is located upstream of the enclosure where the buildings of the mining exploitation are located. In the former Ilba exploitation enclosure are the buildings, silos and water basins (Figure 5) used in former mining activity. By Romanian legislation [53–55], the mines from Ilba were closed.

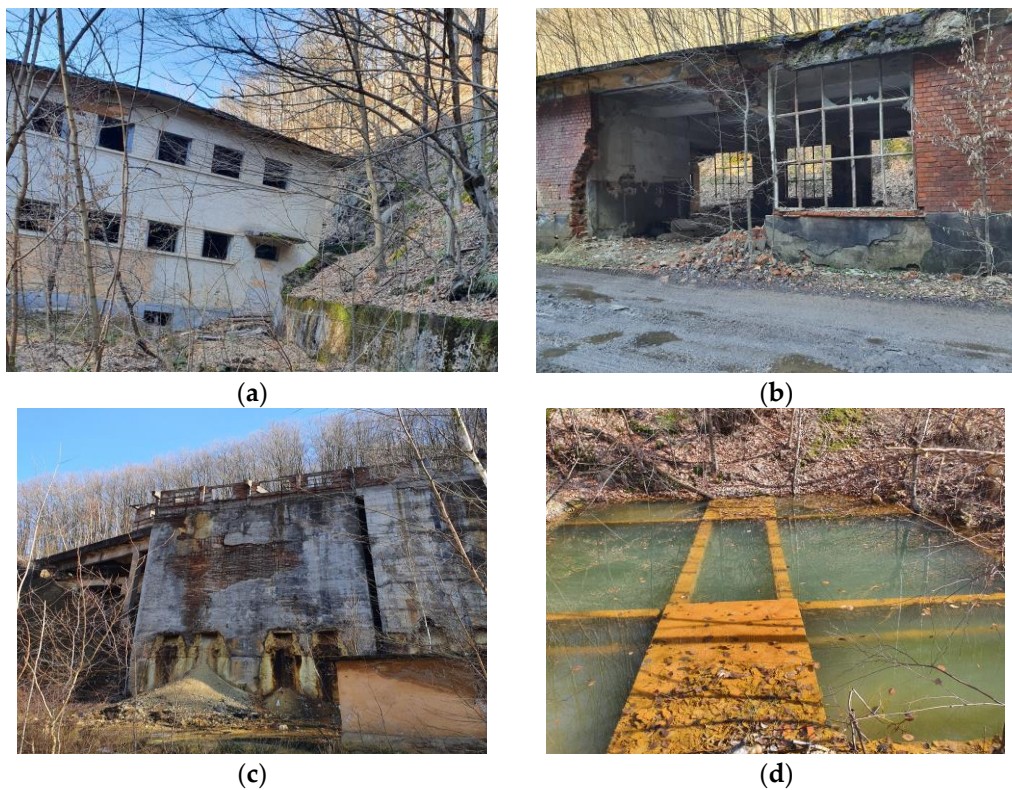

(a)    (b)

(c)    (d)

**Figure 5.** The ruins of the mining exploitation in Ilba, Romania (photos: I.A.P.): (**a**) Ruins of the administrative building; (**b**) Decommissioned building; (**c**) Silos with sterile material; (**d**) Basins with water.

In the Ilba area, some sterile mining dumps (Figure 6) are situated near houses, making their possible instability a risk.

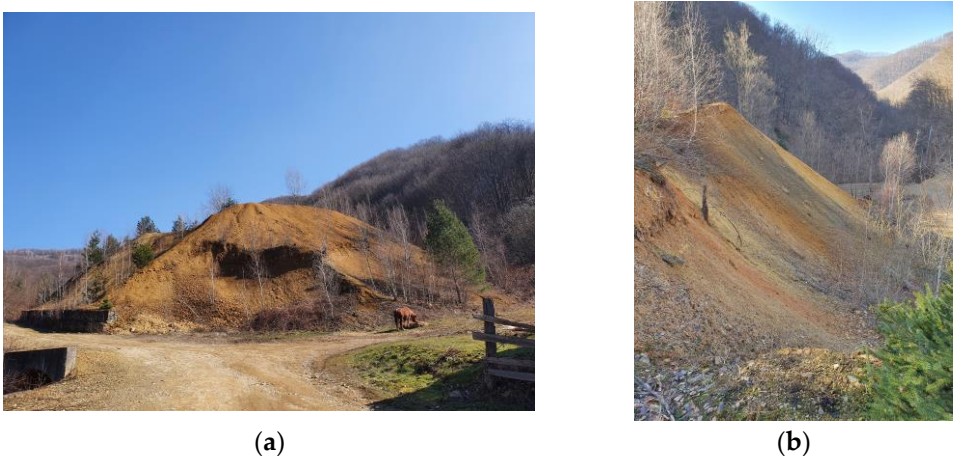

(a)    (b)

**Figure 6.** Sterile dumps, Ilba area, Romania (photos: I.A.P.): (**a**) Sterile dump in the vicinity of the households from which the material slid; (**b**) Sterile dump in the vicinity of road.

The present study will allow us to determine the values of heavy metal concentrations and specific descriptive parameters, such as the pH, the structure and the texture of the Maramures county sterile dumps. An analysis of the compounds and molecules present in the samples and the source of heavy metals in the studied sites will also be conducted.

## 2.2. Sample Collection

Sampling was carried out in accordance with the methodological norms from STAS 7184/1-75 [56]. A total of 27 sterile samples were collected from the Băiuț sterile dump from 9 sampling points at three different depths (A: 0–15 cm, B: 15–30 cm and C: 30–60). One soil sample was collected from a depth of 15–30 cm after removing plant debris from the soil surface.

From the Ilba sterile dump, 10 sterile samples were collected from five sampling points. Two samples were collecting from each point at different depths (A: 0–15 cm and B: 15–30 cm). One soil sample was collected from the Ilba area, located 600 m upstream of the sterile dump from a depth of 0–15 cm after removing plant debris from the soil surface.

The sampling points (Figure 7) were established based on on-site visual inspections and considering the land's topography.

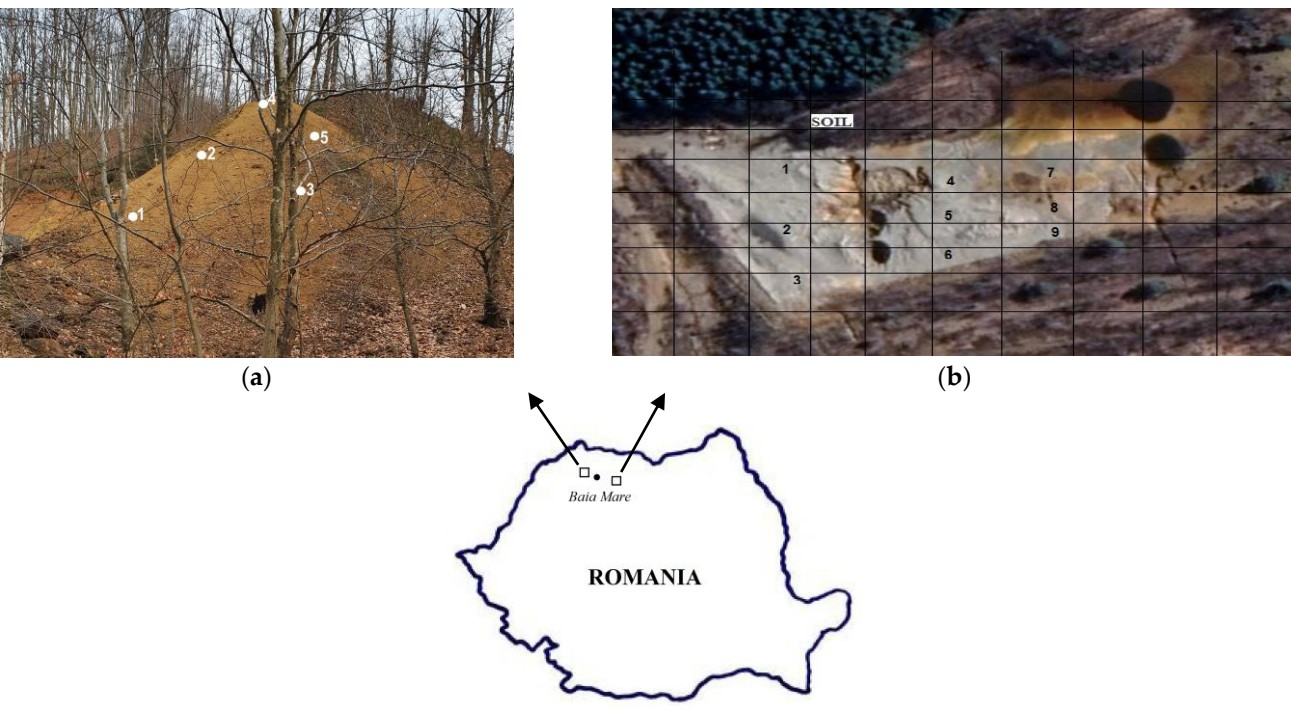

**Figure 7.** Sampling points on the Maramures county mining sites in Romania: (**a**) 1–5: sampling points from Ilba sterile dump; (**b**) 1–9: sampling points from Băiuț site (https://www.google.ro/maps, accessed on 22 March 2021).

## 2.3. Sample Preparation and Analysis

After collection, the samples were dried at room temperature for about 2 weeks using laboratory of Procedures and Remediation Equipment for Soil of the Technical University of Cluj Napoca, Romania. First, the texture of the samples was determined by separating the granulometric fractions using the sieving method with the RETSCH (EP 0642844) AS 200 sieving machine with five sieves (250 μm, 500 μm, 1 mm, 2 mm, 4 mm), according to the standardized determination of granulometry in STAS 1913/5-85 following Romanian legislation [57].

By using a multiparameter Multiline IDS–3430, the pH of the samples was determined according to STAS 7184/13-88 [58] and ISO 10390:2021 [59,60] in an aqueous solution of sterile, respectively, soil with a soil:water mass ratio of 1:10, as follows: 10 g from each collected sample was placed on the 0.1 g accuracy balance. Then, the weighted material was placed in a 100 mL glass recipients and 100 mL of distilled water was added. After homogenization, the mixture was left to rest for 2 h; then, the pH was measured.

The Sekera method [61] was used to determine the structure of the aggregates in each sample. The assessment of the stability of soil aggregates subject to water action was standardized in ISO 10930 from 2012 at an international level [62].

Sample points 1–3 from each site were analyzed by using a Bruker Tracer 5i portable X–ray fluorescence with a 5 kV and 4-watt X–ray source, 8 μm Be window and 8 mm spot collimator. p–XRF is considered an effective, non-destructive method and user-friendly, and can determine the total elemental content of the samples without having to laboriously pre-treat them [63]. X–ray fluorescence can quickly and non-destructively quantify various elements in the soil simultaneously [64].

The molecules from one sample from each sterile dump were analyzed by FT–IR spectrometry using a Nicolet iS50 FT–IR Spectrometer, which measures the absorption spectrum in the mid–IR region (5000–400 $cm^{-1}$). The spectrometer is equipped with a deuterated triglycine sulfate (DTGS) detector and a mercury cadmium telluride (MCT) detector for increased sensitivity and speed. The measurement parameters were: number of sample scans: 100, number of background scans: 100, resolution: 4.000, sample gain: 1.0, optical velocity: 1.8988, aperture: 87.00.

For the extraction of trace elements, each sample was prepared according to legislation SR ISO 11464 [65] regarding soil extractions of trace elements in *aqua regia*. A subsample of 3 g from each collected sample smaller than 150 μm was weighted, placed in glasses and moistened with 0.5–1.0 mL of distilled water. Then, concentrated acids were added: 21 mL hydrochloric acid (37%) and 7 mL nitric acid (65%). After mineralization and disaggregation on the sand bath in a niche, the glasses were cooled and passed through filter paper into 100 mL volumetric flasks.

In compliance with the international standardized method ISO 22036:2008 [66], the concentrations for nine heavy metals were analyzed from the collected samples using an inductively coupled plasma optical emission spectrometer (ICP–OES) Optima 5300 DV (Perkin Elmer, Waltham, MA, USA). The operating conditions used for ICP–OES determination were 1300 W RF power, 15 L $min^{-1}$ Ar plasma support, 2.0 L $min^{-1}$ auxiliary Ar flow, 0.8 L $min^{-1}$ nebulization Ar, and 1.5 mL $min^{-1}$ sample uptake rate, in axial viewing option of the plasma. Then, 7–point linear calibration curves over the range 0–20 mg $L^{-1}$ element were plotted.

## 3. Results

### 3.1. Physical and Chemical Properties of Samples

The granulometry determinations revealed that the texture of the sterile samples collected from the Băiuț sterile dump resulted from ore processing is composed of 32.52% sand mixed with 67.48% dust, and the soil sample of 93.57% sand and 6.43% dust. From the sterile samples collected from ore exploitation in the Ilba mining dump, it was found that the sterile material is made from 83.49% sand and 16.51% dust. The results in the samples collected from the Baiut and Ilba sterile mining dumps are presented as the arithmetic mean for coarse sand (>0.2 mm) and fine sand (0.2–0.02 mm) in each of the sampling points (Figures 8 and 9).

According to the measured pH values in the studied areas, it was found that the predominant pH in both sterile dumps is highly acidic (≤3.5). The soil from the Băiuț area is highly acidic, while it is moderately acidic in the Ilba area (5.1–5.4). The average pH at each point in the sterile samples for the studied areas is presented in Figures 10 and 11.

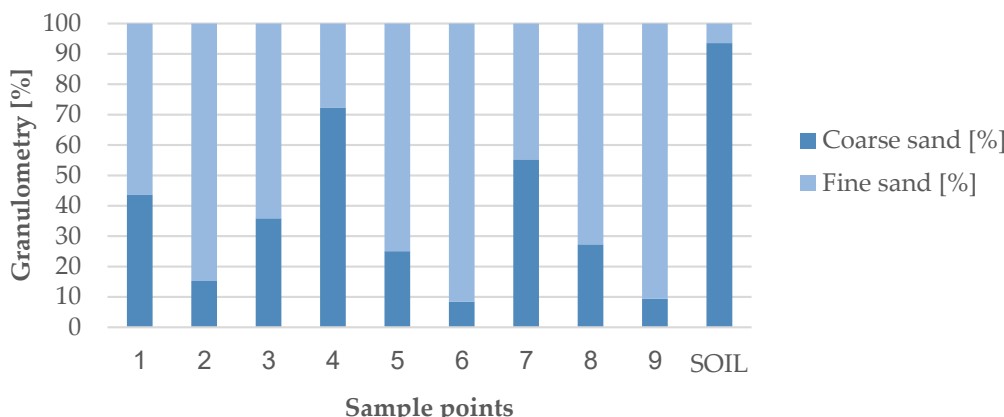

**Figure 8.** Variation in coarse and fine sand in the samples, Băiuț mining area [%].

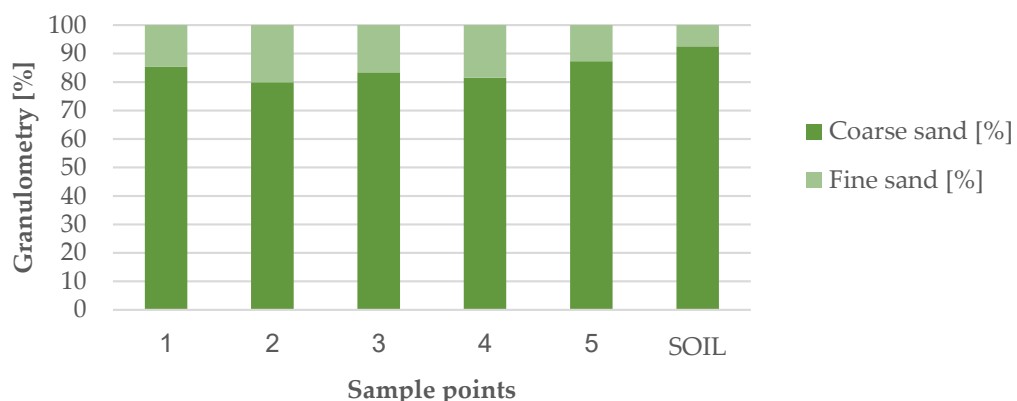

**Figure 9.** Variation in coarse and fine sand in the samples, Ilba mining area [%].

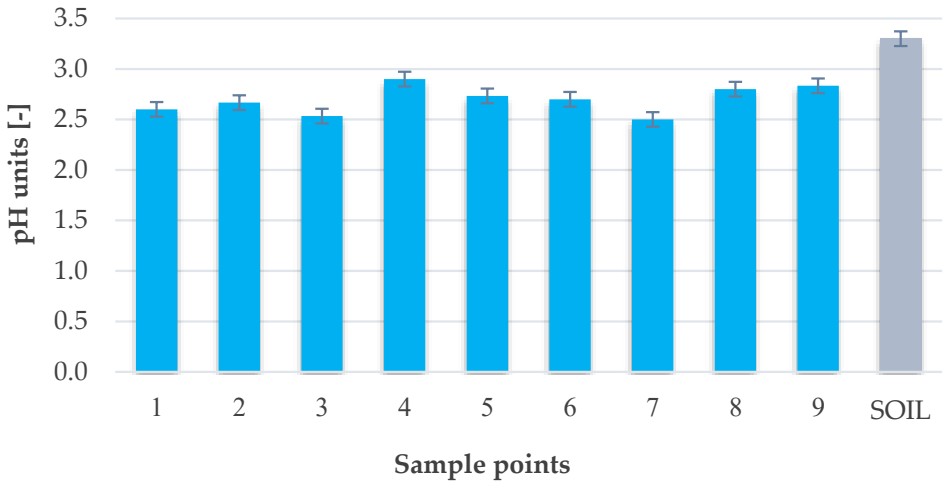

**Figure 10.** Values of pH in the samples, Băiuț mining area.

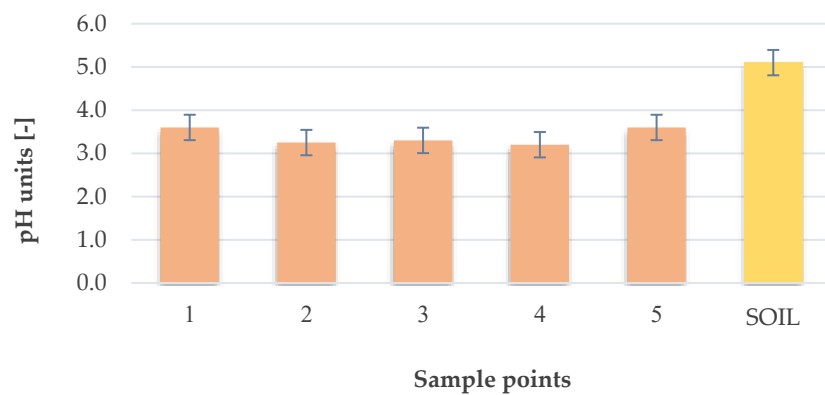

**Figure 11.** Values of pH in the samples, Ilba mining area.

After performing the structure analysis, it was concluded that the sterile material from the Băiuț sterile dump is very poorly structured, and the sterile material from the Ilba dump is poorly structured. The soil sample from the Băiuț area is well structured, and the soil from Ilba area is very well structured.

The FT–IR spectra determinations for the surface samples in point 1A (0–15 cm) collected from each site are presented in Figure 12.

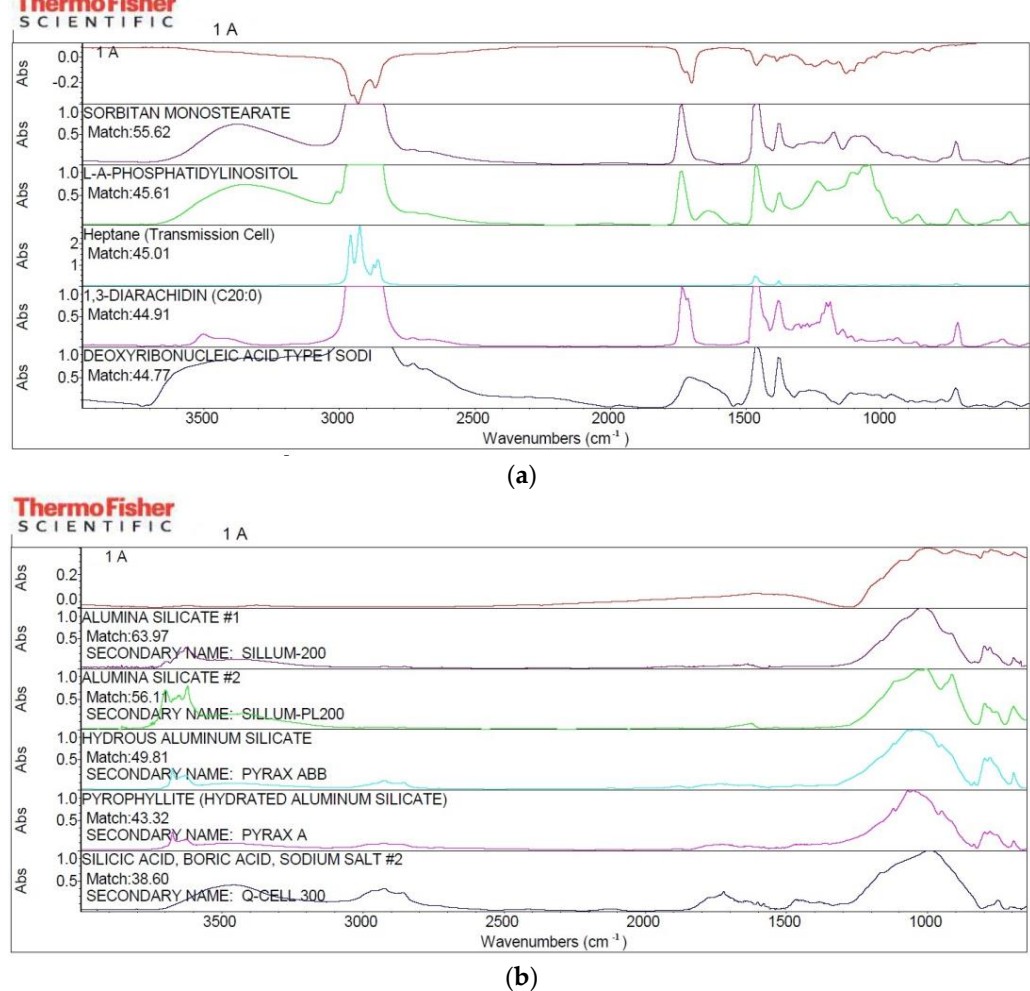

**Figure 12.** FT–IR analysis of sterile material sterile dumps of sample 1A: (**a**) Băiuț sterile dump; (**b**) Ilba sterile dump.

The composition of the samples from each mining dump was investigated for the identification of compounds and analysis of molecules. The molecular structure of the samples is revealed by the absorption properties of the generated absorbance spectrum, indicating at what IR wavelengths each sample absorbs.

The composition of the sterile material collected from three points (1–3) from each of the sterile dumps was analyzed by XRF. The results from the XRF analysis of the samples collected from Băiut sterile dump are presented in Table 1. The XRF results for the analyzed samples collected from the Ilba sterile dump are presented in Table 2. The mean chemical composition of each sample from the sterile dumps is presented in Figure 13.

**Table 1.** Chemical compositions of samples from the Băiuț sterile mining deposit, based on p–XRF measurements.

| Sample | m [%] | | | | | | | | | | | | |
|---|---|---|---|---|---|---|---|---|---|---|---|---|---|
| | $Al_2O_3$ | $SiO_2$ | $P_2O_5$ | S | $K_2O$ | CaO | $TiO_2$ | Mn | Fe | Cu | Zn | As | Pb |
| 1A | 3.09 | 15.11 | 0.16 | 36.41 | 0.29 | 0.76 | 0.13 | <LOD | 36.22 | 0.18 | 0.07 | 0.27 | 0.97 |
| 1B | 0.89 | 18.67 | 0.18 | 26.05 | 0.40 | 0.32 | 0.17 | <LOD | 30.44 | 0.30 | 0.08 | 0.26 | 0.54 |
| 1C | 3.09 | 24.90 | 0.17 | 27.53 | 0.64 | 0.60 | 0.18 | <LOD | 24.30 | 0.15 | 0.07 | 0.27 | 0.18 |
| 2A | 3.56 | 28.82 | 0.16 | 22.34 | 0.63 | 0.22 | 0.21 | <LOD | 19.01 | 0.14 | 0.05 | 0.16 | 0.36 |
| 2B | 5.93 | 45.85 | 0.14 | 2.65 | 1.10 | 0.18 | 0.22 | 0.017 | 5.24 | 0.02 | 0.03 | 0.11 | 0.10 |
| 2C | 5.97 | 47.93 | 0.13 | 1.04 | 0.80 | 0.16 | 0.10 | 0.025 | 3.22 | 0.02 | 0.04 | 0.06 | 0.04 |
| 3A | 0.47 | 17.44 | 0.17 | 30.15 | 0.45 | 0.48 | 0.20 | <LOD | 32.49 | 0.10 | 0.03 | 0.24 | 0.53 |
| 3B | 3.67 | 26.52 | 0.17 | 25.63 | 0.79 | 0.60 | 0.24 | <LOD | 21.25 | 0.11 | 0.09 | 0.43 | 0.16 |
| 3C | 6.12 | 43.42 | 0.14 | 2.25 | 0.85 | 0.19 | 0.17 | 0.024 | 3.92 | 0.05 | 0.05 | 0.18 | 0.03 |

LOD = 0.005% for Mn by p-XRF.

**Table 2.** Chemical compositions of samples from the Ilba sterile mining deposit, based on p−XRF measurements.

| Sample | m [%] | | | | | | | | | | | | |
|---|---|---|---|---|---|---|---|---|---|---|---|---|---|
| | $Al_2O_3$ | $SiO_2$ | $P_2O_5$ | S | $K_2O$ | CaO | $TiO_2$ | Mn | Fe | Cu | Zn | As | Pb |
| 1A | 9.89 | 41.85 | 0.15 | 1.82 | 3.11 | 0.24 | 0.44 | 0.027 | 6.10 | 0.06 | 0.013 | 0.022 | 0.22 |
| 1B | 9.69 | 41.10 | 0.15 | 2.03 | 3.15 | 0.23 | 0.45 | 0.033 | 7.42 | 0.03 | 0.014 | 0.027 | 0.16 |
| 2A | 8.94 | 39.89 | 0.16 | 3.59 | 2.96 | 0.23 | 0.33 | 0.014 | 6.35 | 0.03 | 0.013 | 0.022 | 0.22 |
| 2B | 8.62 | 38.50 | 0.14 | 3.75 | 3.06 | 0.23 | 0.34 | 0.014 | 7.20 | 0.04 | 0.012 | 0.024 | 0.16 |
| 3A | 10.50 | 40.85 | 0.16 | 1.85 | 3.21 | 0.23 | 0.49 | 0.013 | 5.12 | 0.04 | 0.017 | 0.027 | 0.28 |
| 3B | 10.31 | 39.31 | 0.14 | 2.08 | 2.84 | 0.23 | 0.34 | 0.006 | 8.19 | 0.03 | 0.017 | 0.054 | 0.355 |

The sterile material resulting from the exploration and exploitation of ore in Ilba area has a 2.6 times higher average percentage of aluminum oxides, 1.3 times higher average percentage of silica and 4.6 times higher average potassium oxide than the material from the Băiuț area. Regarding the average percentage of the main trace heavy metals, the sterile dump resulting from the processing of ore in the Băiuț area has a 7.52 times higher percentage for As, 4 times higher percentage for Zn, 3.16 times higher percentage for Cu, 1.4 times higher percentage for Pb, and 2.9 times higher percentage for Fe than the sterile from Ilba area. The percentage of Mn in the Ilba sterile dump is 2.47 times higher than in the Băiuț dump.

### 3.2. Heavy Metals

The concentrations of heavy metals (Cd, Cr, Cu, Mn, Ni, Fe, Pb, Zn, Co) measured with ICP-OES for each sample collected from the Băiuț area and the Ilba area are presented in Tables 3 and 4, respectively.

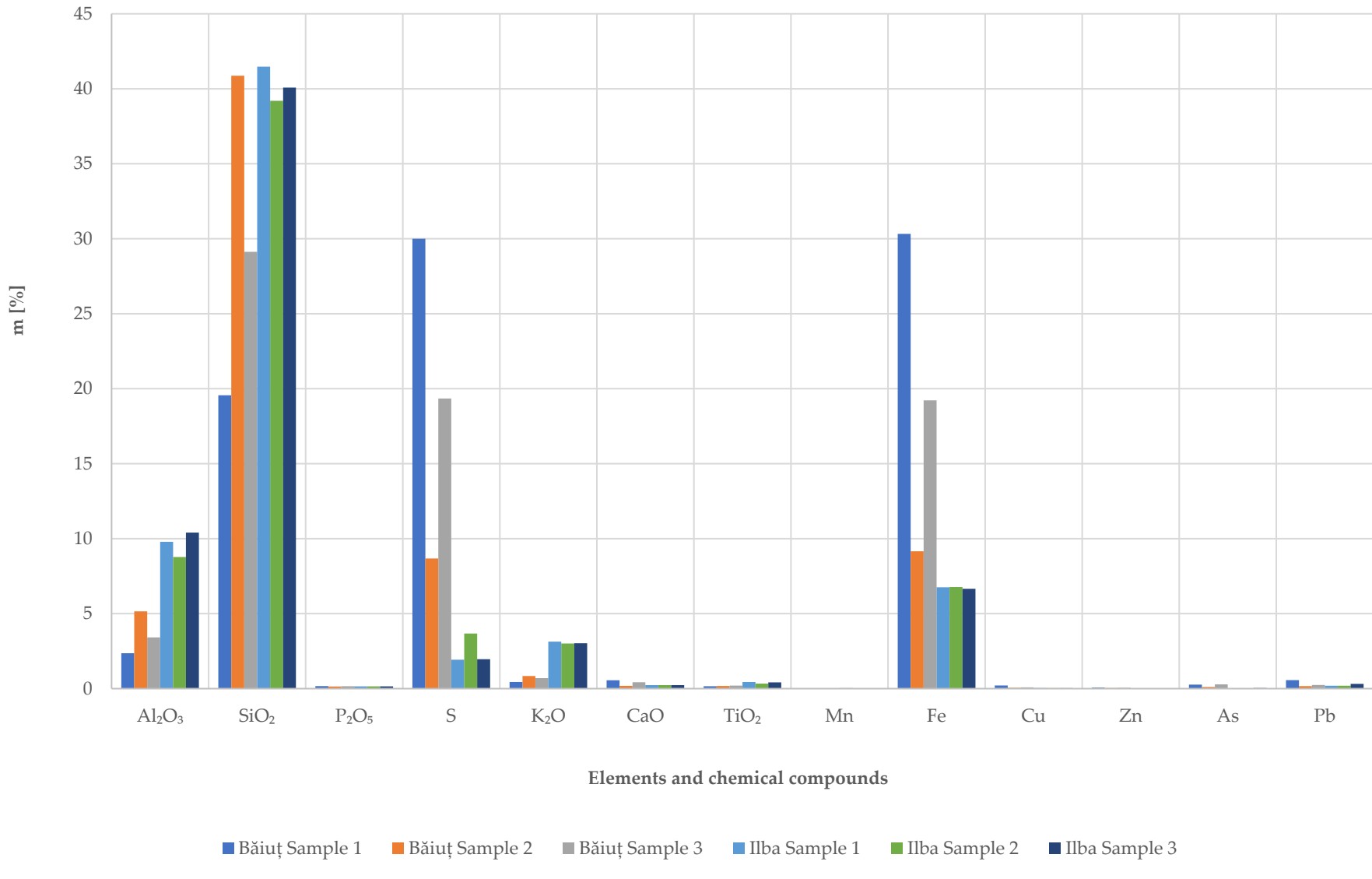

**Figure 13.** Mean chemical compositions of samples from the sterile mining dumps, based on p−XRF measurements.

**Table 3.** The results of the determinations of the heavy metals concentrations in the samples collected from the Băiuț mining area.

| Sample | Metal Concentration [mg kg$^{-1}$] | | | | | | | | |
|---|---|---|---|---|---|---|---|---|---|
| | **Cd** | **Cr** | **Zn** | **Cu** | **Mn** | **Pb** | **Fe** | **Ni** | **Co** |
| 1A | 4.0 | 3.4 | 386 | 1170 | 10.1 | 82.3 | 7990 | 41.0 | 105 |
| 1B | 13 | 4.6 | 521 | 2160 | 9.27 | 262 | 8000 | 38.1 | 101 |
| 1C | <LOD | 6.0 | 404 | 1066 | 11.4 | 53.9 | 8183 | 32.6 | 73.3 |
| 2A | 1.0 | 8.6 | 245 | 478 | <LOD | 53.3 | 8413 | 22.3 | 62.2 |
| 2B | 16 | 8.3 | 213 | 330 | 15.5 | 195 | 8173 | 18.7 | 40.3 |
| 2C | <LOD | 7.2 | 242 | 242 | 62.4 | 128 | 6340 | 13.1 | 13.1 |
| 3A | 9.2 | 1.1 | 235 | 665 | 0.40 | 1393 | 8133 | 44.2 | 104 |
| 3B | 45 | 2.2 | 431 | 738 | 7.6 | 658 | 8460 | 57.0 | 89.4 |
| 3C | 45 | 13 | 318 | 401 | 63.8 | 187 | 7680 | 13.5 | 12.4 |
| 4A | 10 | 4.5 | 184 | 325 | 32.5 | 521 | 7943 | 23.0 | 74.0 |
| 4B | 7.3 | 26 | 295 | 833 | 417 | 681 | 7257 | 24.6 | 21.0 |
| 4C | 14 | 36 | 673 | 1105 | 1197 | 585 | 7240 | 45.0 | 34.3 |
| 5A | 6.1 | 1.0 | 230 | 390 | 1.4 | 242 | 8547 | 25.9 | 94.0 |
| 5B | 9.0 | 2.0 | 348 | 1115 | 5.5 | 969 | 8540 | 36.6 | 111 |
| 5C | 6.2 | 5.5 | 69.5 | 709 | 29 | 2494 | 7700 | 1.20 | 0.23 |
| 6A | 5.3 | 0.2 | 180 | 127 | <LOD | 656 | 8383 | 25.5 | 98.0 |
| 6B | 6.0 | 0.7 | 273 | 303 | <LOD | 794 | 8510 | 29.0 | 106 |
| 6C | 5.7 | 4.7 | 118 | 162 | 29.6 | 562 | 7730 | 10.0 | 28.9 |
| 7A | 7.3 | 4.8 | 157 | 838 | 32.9 | 361 | 7913 | 17.9 | 30.3 |
| 7B | 9.0 | 15 | 119 | 155 | 56.6 | 482 | 7523 | 4.66 | 3.16 |
| 7C | 9.1 | 14 | 108 | 134 | 52.5 | 471 | 7470 | 5.23 | 5.20 |
| 8A | 7.0 | 2.8 | 215 | 815 | 0.11 | 595 | 8293 | 27.0 | 46.5 |
| 8B | <LOD | 6.6 | 38.1 | 47.0 | 6.0 | 325 | 6227 | 1.93 | 1.63 |
| 8C | 14 | 13 | 130 | 179 | 17.0 | 302 | 7430 | 10.1 | 11.7 |
| 9A | 9.0 | 5.9 | 158 | 965 | 96.3 | 2101 | 7603 | 8.93 | 9.00 |
| 9B | 10 | 9.5 | 134 | 211 | 72.3 | 605 | 7327 | 7.60 | 2.50 |
| 9C | 5.0 | 11 | 69.1 | 133 | 26.7 | 350 | 6933 | 6.46 | 4.43 |
| SOIL | <LOD | 30 | 94,3 | 51.1 | 371 | 121 | 7150 | 21.9 | 9.33 |

LOD = 0.10 mg kg$^{-1}$ for Cd and Mn by ICP−OES.

**Table 4.** The results of the determinations of the heavy metals concentrations in the material collected from the Ilba mining area.

| Sample | Metal Concentration [mg kg$^{-1}$] | | | | | | | | |
|---|---|---|---|---|---|---|---|---|---|
| | **Cd** | **Cr** | **Zn** | **Cu** | **Mn** | **Pb** | **Fe** | **Ni** | **Co** |
| 1A | <LOD | 9.4 | 82 | 1113 | 113 | 1224 | 7727 | 3.5 | 0.6 |
| 1B | <LOD | 6.0 | 128 | 738 | 173 | 756.7 | 6900 | 3.2 | 0.4 |
| 2A | <LOD | 3.0 | 57 | 522 | 27.9 | 808.3 | 7630 | 0.9 | 0.5 |
| 2B | <LOD | 3.5 | 81 | 455 | 91.9 | 823.7 | 7897 | 0.9 | 0.3 |
| 3A | 3.0 | 7.1 | 65 | 533 | 16.2 | 1816 | 7327 | 2.0 | 0.2 |
| 3B | 14 | 7.6 | 270 | 409 | 15.7 | 448,3 | 7783 | 13 | 34 |
| 4A | 3.1 | 3.8 | 72 | 738 | 26.4 | 1814 | 7803 | 1.0 | 0.2 |
| 4B | 1.2 | 4.0 | 64 | 653 | 25.2 | 1806 | 8010 | 1.0 | 0.2 |
| 5A | 2.0 | 5.9 | 92 | 961 | 61.8 | 2191 | 7687 | 1.2 | 0.3 |
| 5B | 1.3 | 5.9 | 68 | 467 | 44.4 | 1662 | 7663 | 1.1 | 0.3 |
| SOIL | 1.0 | 1.0 | 96 | 61 | 651 | 208.7 | 3430 | 1.4 | 7.8 |

LOD = 0.10 mg kg$^{-1}$ for Cd by ICP−OES.

The mean concentrations of the ICP−OES determinations for the sterile mining dumps after calculating the mean concentration in each point for each trace element are presented in Figure 14.

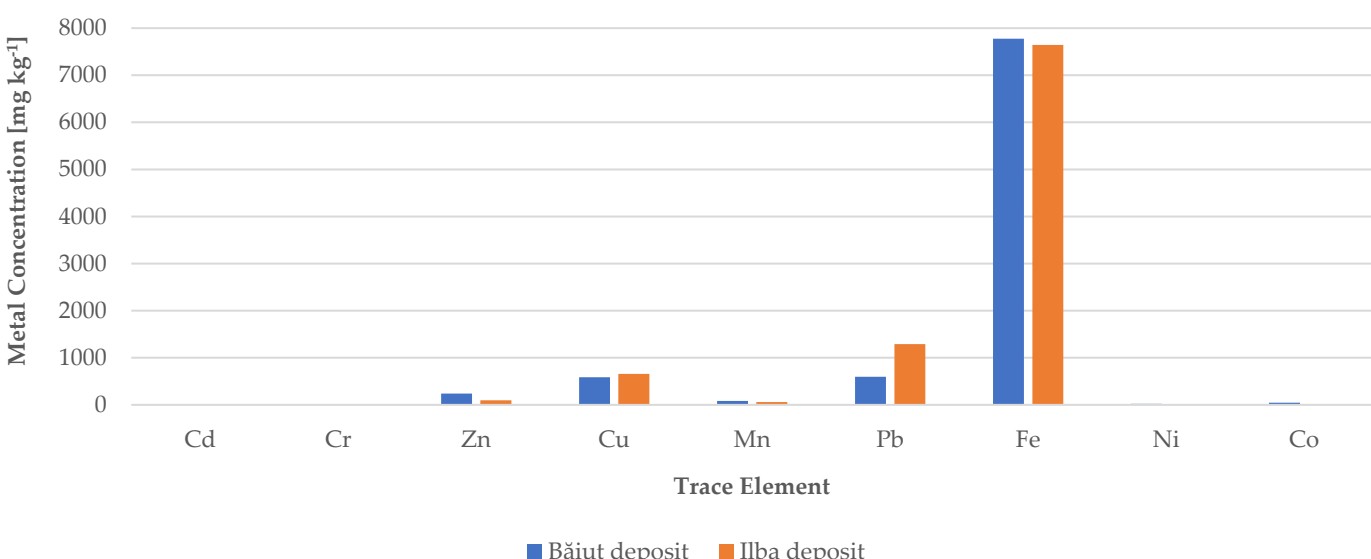

**Figure 14.** Mean trace element concentrations in the sterile mining dumps, based on ICP−OES determinations.

Overall, similar concentrations of Fe are found in both deposits; Pb and Cu is higher in the Ilba sterile dump, and Zn is higher in the Băiuț sterile dump (Figure 14).

## 4. Discussions

The mineralogical composition of the sterile mining material and its size distribution depends on the alteration processes caused by climatic agents and age [67,68]. The reactivity of the sterile material depends on chemical composition, grain size, crystallinity, texture and grain heterogeneity [69]. The main features of materials from mining dumps are excessive texture (too sandy or too clayey), excessive skeletal content, low humus content, low contents of nitrogen, phosphorus, potassium, micronutrients, poor aero-hydric regime and reduced biological activity [70]. When mining sites are exposed to weathering processes over time, a young type of soil known as mine soil develops, which has very high concentrations of heavy metals, no functional ecosystems, is devoid of vegetation for extended periods, and has poor water holding capacity, low organic matter content, nutrient deficiency, and low microbial activity [12].

The sterile dumps resulting from the exploitation and processing of metal ores (Cu, Pb, Zn, Au-Ag, Fe) are rich in sulfur minerals, resulting in the exposure of metal sulfides to the action of the atmospheric oxygen. Acidification of sterile dumps is given by the oxidation of sulfides, mainly pyrite, and is often linked with a reduction in pH, the release of heavy metals and the precipitation of Fe (III) −(oxy) hydroxides and Fe (III) −hydroxy sulfates [3]. Local environmental concerns are represented by mobile harmful elements released from the sterile mining dumps that were not bound from primary sulfides [71].

The physicochemical characterization of the sterile material is critical, mainly if it is deposited under subaerial conditions where it is subjected to precipitation leaching of heavy metals that can be transported to the surrounding area by acid mine drainage and oxidation processes [3,72]. According to XRF determinations, oxides are the primary sources of the heavy metals in the studied sterile mining dumps (Tables 3 and 4). Due to precipitation, the chemical−bacterial solubilization of metal sulfides contained in sterile dumps produces solutions loaded with copper and other heavy metals, which enter by infiltration in the field and groundwater or reach the local and regional hydrographic system [73].

The average concentrations of heavy metals in the samples of sterile material from ore processing analyzed with ICP−OES from the Băiuț area were 10.1 mg kg$^{-1}$ Cd, 8.15 mg kg$^{-1}$ Cr, 585 mg kg$^{-1}$ Cu, 83.4 mg kg$^{-1}$ Mn, 7775 mg kg$^{-1}$ Fe, 21.9 mg kg$^{-1}$ Ni, 596 mg kg$^{-1}$ Pb, 240 mg kg$^{-1}$ Zn and 47.5 mg kg$^{-1}$ Co. In the sterile dump from mineral ex-

ploitation in the Ilba area, according to the concentrations of the analyzed heavy metals, the average concentration for each element is 2.55 mg kg$^{-1}$ Cd, 5.63 mg kg$^{-1}$ Cr, 658 mg kg$^{-1}$ Cu, 59.5 mg kg$^{-1}$ Mn, 7643 mg kg$^{-1}$ Fe, 2.78 mg kg$^{-1}$ Ni, 1335 mg kg$^{-1}$ Pb, 98.0 mg kg$^{-1}$ Zn and 3.71 mg kg$^{-1}$ Co. The intervention thresholds for soils with less sensitive use for Cd, Cr, Zn, Cu, Mn, Pb, Ni and Co are 10 mg kg$^{-1}$, 600 mg kg$^{-1}$, 1500 mg kg$^{-1}$, 500 mg kg$^{-1}$, 4000 mg kg$^{-1}$, 1000 mg kg$^{-1}$, 500 mg kg$^{-1}$ and 250 mg kg$^{-1}$, respectively, according to the Romanian legislation of the Ministry of Water, Forests and Environmental Protection regarding the assessment of environmental pollution [74]. Therefore, the average concentrations of the samples collected from the Baiuț area exceed the intervention threshold for land with less sensitive use for two of the determined trace elements (Cd and Cu). In the Ilba area, the intervention threshold for land with less sensitive use was exceeded for the average concentrations of Cu and Pb.

Similar ICP−OES determinations were carried out in a waste rock mining heap from Jiangxi Province, China [4], and ICP-MS and ICP-ES in three mining waste deposits from Slovenia (the Mežica, Litija and Pleše areas) [3]. In mine soils from NW Spain, the highest heavy metal concentration analyzed by AAS showed values 77 times higher for Zn, 2.4 times higher for Pb, 2.1 times higher for Cd, and 13.5 lower for Cu, 1.4 times lower for Ni, 1.7 lower for Cr and 4.1 times lower for Co, than in the analyzed sites from Maramures county [2]. In a pyritic tailing impoundment located in the mining area of the Iberian Pyrite Belt (Minas de Ríotinto-Zarandas), SW Spain, the mean concentrations were 1.6 times lower for Cu, 2.4 limes lower for Pb and 1.1 times higher for Zn in the non-topsoiled sector of Zarandas dam and 2.1 times lower for Cu, 4 times lower for Pb and 1.2 times lower for Zn in the topsoiled sector than the mean concentrations of sites studied in the county of Maramures [75].

In the area of Ponce Enríquez in Azuay province, Southern Ecuador, where illegal mining activities generate numerous sterile dumps, wastes, and tailings, the mean concentrations of heavy metals in soils from patios, gardens, and public spaces were 1.5 times lower for Cd, 11 times higher for Cr, 3.7 times lower for Cu, 16.8 times higher for Ni, 71.4 times lower for Pb and 1.4 times lower for Zn than the sterile mining dumps in Maramures county [22]. In tailings from the semi-arid Iron King Mine and Humboldt Smelter Superfund site in central Arizona (Dewey-Humboldt, AZ, USA), the concentration for heavy metals was 2.3 higher for Pb and 15 times higher for Zn than the mean concentrations of the Maramures sites [67].

The mean concentrations determined in the samples collected from the emplacement of a former sterile dump resulting from the processing of ore in Maramures county (Romania) in the region of Baia Mare were 39.8 times higher for Cd, 5.8 times higher for Cr, 4.8 times higher for Zn, 3.9 times higher for Cu, 1.5 times higher for Mn, 1.92 times lower for Pb and 2.2 times higher for Co than the mean concentration of the sites studied in this research [76].

The contamination of sterile mining sites is not visible because the contamination is located in the underground environment for several years or even decades and is discovered once it has reached a specific target, such as water supply [1]. In order to limit the dispersion of sterile material in groundwater, rivers, lakes and wind and to avoid contamination of drinking water and food chains, the proactive isolation of the dumps is required. The reclamation strategies that can be used to limit the oxidation of sulfide minerals in tailings that generate acid mine drainage [77,78] include creating a high degree of saturation by keeping the water table at a depth less than the air entry value using a monolayer cover combined with an elevated water table [79,80], or using a cover with capillary barrier effects, which involves the retention of water by a layer of fine-grained material placed between two coarse material layers [79].

The implementation of national regulatory procedures to ensure compliance with environmental protection requirements to solve the ecological problems resulting from the mining industry, such as unmonitored and abandoned mining sites and drainage of the acid mines, continues [41,81]. There is an evident need for effective treatment strategies for abandoned sterile mining dumps, and there are multiple ways to manage the sterile

dumps, which were left exposed to environmental factors [82,83]. The remediation methods of the sterile dumps are physical, chemical and biological [81,84–87]. Biological remediation strategies for heavy metal removal from soil include phytoremediation, microbial remediation, and animal remediation [88,89]. The study on the ecological restoration of the Lviv-Volyn mining basin in western Ukraine by phytomelioration showed that when climatic humidity, shadowing in cenosis, and soil moisture decreased, thermal regime, continentality, soil pH, and salt content increased [90].

The bioavailability and toxicity of heavy metals in sterile mining dumps can be reduced by using microorganisms to adsorb, precipitate, oxidize or reduce heavy metals [91,92]. Novel biological remediation strategies that use microbial metabolism (aerobic or anaerobic) to stimulate favorable biotic and abiotic reactions that transform toxic compounds into innocuous substances, with microbial sulfate reduction, are particularly attractive [93]. Some soil-dwelling animals (larvae, earthworms, etc.) can absorb heavy metals from the soil, as it has been shown that earthworm activity and secretion at low soil Cu concentrations may facilitate Cu uptake by *Lolium perenne* [94].

Mining waste products, such as sterile material resulting from the removal of rocks for the exploitation of ore, minerals, and chemicals, and water used in mineral extraction treatments, can have a transformative effect on the environment [95,96]. The consequences of polluted sites can include the depletion of water resources, flora and fauna, and diseases, rendering any economic activity unsustainable (agriculture, livestock, fishing) [1].

## 5. Conclusions

This study focused on two mining sites, which have unique features such as high heavy metal concentrations with oxides as the primary sources and an acidic pH, making remediation extremely challenging. The predominant pH in both sterile dumps is highly acidic ($\leq 3.5$). The texture of the sterile dump ore processing site at Băiuț is composed of 32.52% sand and 67.48% dust and the sterile dump from ore exploitation at Ilba is composed of 83.49% sand and 16.51% dust. Both sterile dump from Băiuț and the sterile dump from Ilba are very poorly structured.

According to the terms of Romanian legislation regarding the assessment of environmental pollution, the sterile samples taken from the Baiuț area were found to exceed the intervention threshold for Cd and Cu concentrations for land with less sensitive use, while samples taken from the Ilba area were found to exceed the intervention threshold for Cu and Pb concentrations. On the other hand, concentration values for heavy metals exceeding the intervention threshold for less sensitive soil uses were not found in the soil samples. Thus, some pollution control measures are required. This study provides new and actual information regarding the state of contamination of the studied sites. These data are of value for the local restoration of the affected areas. The high concentrations of heavy metals determined in the studied areas necessitate site remediations to decrease the risk of contamination in neighboring regions and alleviate the possible human health risks and agricultural output from surface or groundwater pollution, aeolian dispersion, water erosion, or absorption by plants and bioaccumulation in food chains.

Non-conventional remediation methods could be used to remediate the studied areas and reduce the risk of collateral pollution these mining wastes dumps can produce. Further research regarding the spatial and temporal extent of degradation status in these mining sites should be conducted in the future. This study recommends that regular monitoring of heavy metals in these areas be carried out to guard against pollution.

**Author Contributions:** Conceptualization, I.A.P. and V.M.; methodology, I.A.P. and V.M.; investigation, I.A.P. and M.Ș.; resources, V.M. and M.Ș.; writing-original draft preparation, I.A.P.; writing-review and editing, I.A.P., V.M. and M.Ș. All authors have read and agreed to the published version of the manuscript.

**Funding:** This paper was financially supported by the Project "Entrepreneurial competences and excellence research in doctoral and postdoctoral programs-ANTREDOC"; project co-funded by the European Social Fund financing agreement no. 56437/24.07.2019.

**Institutional Review Board Statement:** Not applicable.

**Informed Consent Statement:** Not applicable.

**Data Availability Statement:** Not applicable.

**Conflicts of Interest:** The authors declare no conflict of interest.

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
