# Peer review of "Investigation of Sterile Mining Dumps Resulting from Ore Exploitation and Processing in Maramures County, Romania"

_land, doi:10.3390/land12020445_

Round 1

Reviewer 1 Report

Manuscript entitled “Investigation of sterile mining dumps resulted from ore exploitation and processing from Maramures county (Romania)” is a good attempt and subject matter is a great importance to in present days land degradation and deteriorating soil health to achieve sustainable development goal. However Manuscript is lacking proper methodology and proper research orientation goal. Some of the major limitation of the manuscript is

1. Manuscript just compares two sterile mining dump sites and their properties. It is a good report but cannot be research manuscripts which has some hypothesis why research work has been carried out, proper methodology to achieve that, significant research results and proper discussion why its is happening. Manuscript is lack in all those aspects.

2. No statistical survey design and statistical comparison have been made.

3. No spatial or temporal analysis have been done. So that we can get information what its spatial and temporal extent of degradation status in those mining sites.

4. No proper discussion have been done. Whats steps need to be done to control degradation status and what extent government is taking steps to control degradation.

Therefore manuscript can not be accept in its present form and advice to incorporate all these suggestion and present in a proper research manuscript format.

Reviewer 2 Report

The authors describe the historical contamination sites but don’t have any analysis on structure us layers type protection infrastructure flooding control infrastructures. The descriptive functioning of the mention two sites is not mentioning the precipitation received and the debts of effluent. Also, they mention more times the structures of industrial mining infrastructure are not appropriate to actual environmental standards, but they don’t make specific analysis of structure, just mention the content of potential chemical pollutants concentration on the ore sterile dumps.

They need to reorganize the point 2.2 mining risk factors associated to the study areas, they mention the presence of ravens and erosion but they don’t have any quantitative analysis to reflect the potential risk based on their sampling strategy and results of pollutants concentration.  The effluent “responsible” to river pollution and atmospheric erosion of ore sterile dumps are not covered by research sampling. The spatial interpolation of the concentration observed by sampling point maybe can explain the areas with potential exposure to identified environmental risk.  

The conclusion are general affirmation but not sustained by current research.  

Reviewer 3 Report

The present manuscript describes the investigation into the characteristics of a sterile mining dump from the exploitation of ore and the characteristics of a sterile dump from ore processing by determining their physicochemical properties and the concentrations of heavy metals to propose remediation techniques for the affected areas. A manuscript has a practical application and also provides important theoretical for the next studies as the obtained results represent the starting point to establish the necessity to remediate the affected areas and to propose several remediation techniques.

The paper can be accepted for publication after providing the corrections mentioned below.

Point 1. There are too many Keywords.

Point 2. In the Introduction section, an enhanced literature review is required. It will be great if the authors show some description in context – Why it is important to conduct this study?

Point 3. Why do not show in your paper that your research is important not only for the Romania, but it is a wide problem in Europe and other countries. In case to provide enhanced literature review (according to Point 2) please consider the suggested research (comes from, Kosovo and Ukraine) in your paper when enhancing the introduction section. I believe they are worth considering in your paper.

Sadiku, M., Kadriu, S., Kelmendi, M., & Latifi, L. (2021). Impact of Artana mine on heavy metal pollution of the Marec river in Kosovo. Mining of Mineral Deposits, 15(2), 18-24. https://doi.org/10.33271/mining15.02.018

Kadriu, S., Sadiku, M., Kelmendi, M., & Sadriu, E. (2020). Studying the heavy metals concentration in discharged water from the Trepça Mine and flotation, Kosovo. Mining of Mineral Deposits, 14(4), 47-52. https://doi.org/10.33271/mining14.04.047

Skrobala, V., Popovych, V., Tyndyk, O., & Voloshchyshyn, A. (2022). Chemical pollution peculiarities of the Nadiya mine rock dumps in the Chervonohrad Mining District, Ukraine. Mining of Mineral Deposits, 16(4), 71-79. https://doi.org/10.33271/mining16.04.071

Moreover, you are welcome to use more recently published paper in your research.

Point 4. The tasks of the research should be highlighted at the end of the Introduction section.

Point 5. Try do not use references in the Results section because authors results only must be given in this section. At the same time, you can compare your results with adding references.

Point 6. Please provide a short description of further research.

Point 7. The novelty of the paper must be highlighted in the conclusions section.

Point 8. The content of the manuscript is similar to that of a case study. The knowledge contained here may be useful for engineers, students, and scientists, searching for any knowledge related to mining engineering, which is the most important value of the manuscript.

I will recommend your paper for publication after careful revision.

Round 2

Reviewer 2 Report

The analysis of potential environmental risk (ERA) is based on spatial and temporal assessment of current infrastructure and processes. It will better to have in mind the proportions of remaining industrial buildings (not just photos also we need maps to understand extent), industrial waste deposits (volumes, stabilities, presence of erosion, percolation estimates) and effluents (debts, accumulation basin, surface or ground water connections).  After ERA we can consider alternatives of the restoration methods/actions to be consolidated into strategic action plan of restauration. 

Reviewer 3 Report

Dear authors,

I am more than satisfied with the corrections provided by you. This study is an important contribution to the research field.

Congratulations to the authors.